# Molasses-Based Block Supplements for Cattle Fed Endophyte-Infected Tall Fescue (*Festuca arundinacea*) Seed: Effects on Growth Performance, Circulating Biomarkers, Heat Stress, and Coccygeal Artery Diameter

**DOI:** 10.3390/ani15050717

**Published:** 2025-03-03

**Authors:** Luis F. B. B. Feitoza, Brad J. White, James S. Drouillard

**Affiliations:** 1Beef Cattle Institute, Kansas State University, Manhattan, KS 66506, USA; lffeitoza@vet.k-state.edu (L.F.B.B.F.); bwhite@vet.k-state.edu (B.J.W.); 2Department of Animal Sciences and Industry, Kansas State University, Manhattan, KS 66506, USA

**Keywords:** heat stress, molasses-based blocks, tall fescue, thermography, vasoconstriction

## Abstract

Tall fescue is a widely used forage for cattle, but it can be infected with a microorganism that produces toxic compounds called ergot alkaloids. These toxins reduce blood flow, causing heat stress, poor weight gain, and even tissue damage in cattle. Currently, there are limited options to prevent these negative effects besides removing animals from infected pastures. This study tested whether molasses-based block supplements could help cattle tolerate ergot alkaloids by improving blood circulation and reducing heat stress. Cattle were given different types of block supplements, including those containing menthol or capsaicin, compounds known for their ability to influence blood flow. The results showed that cattle consuming crude protein-enriched blocks had better growth performance and improved blood circulation compared to those that did not receive supplements. In particular, cattle that consumed these supplements had larger blood vessel diameters and better heat regulation. These findings suggest that molasses-based block supplements may offer a practical and accessible way to mitigate the harmful effects of ergot alkaloids in cattle grazing on tall fescue pastures.

## 1. Introduction

The occurrence of ergot toxicosis (i.e., fescue foot or gangrenous ergotism) in grazing animals has been reported throughout the world [1]. Ergot alkaloid mycotoxins are secondary metabolites of fungi of the genera *Claviceps* and *Epichloe* spp. [1,2,3]. These fungi can infect grazed forages such as tall fescue (*Festuca arundinacea*), leading to the clinical condition of fescue toxicosis, characterized by reduced blood flow to peripheral tissues. Altered blood flow is the consequence of ergot alkaloid-induced vasoconstriction by adrenergic and serotonin agonistic effects. Accompanying symptoms of toxicosis include abdominal fat necrosis [4]; hyperthermia; decreases in feed intake, body weight gain, and milk production [5]; lameness; and ear tip and tail tip necrosis [6], thus compromising animal well-being and performance. Moreover, the lack of alternative intervention methods besides removing the animals from the source of ergot alkaloids urges for alternative methods to mitigate this problem.

Menthol is a multi-purpose compound widely used as a flavoring agent, analgesic, antiseptic, and vasoactive agonist [7]. It acts as an agonist to transient receptor potential cation channel subfamily Melastatin-related member 8 (TRPM8), a cold-sensitive ion channel involved in thermoregulation and vasodilation [8]. TRPM8 receptors are expressed in various tissues, including arterial smooth muscle, where activation leads to calcium influx and subsequent vasodilation. This effect is partially mediated by an increase in nitric oxide (NO) bioavailability, which facilitates endothelium-dependent relaxation of blood vessels [9]. Menthol has been reported to induce cutaneous vasodilation, suggesting its potential role in improving blood flow under heat-stressed conditions in livestock [9].

Capsaicin, the active compound in chili peppers, binds to transient receptor potential vanilloid 1 (TRPV1) receptors, which play a crucial role in thermoregulation and vascular function [10]. TRPV1 activation by capsaicin has been linked to enhanced expression and activation of endothelial nitric oxide synthase (eNOS), leading to increased nitric oxide production and vasodilation [11]. This mechanism contributes to reduced vascular resistance, which may counteract the vasoconstrictive effects of ergot alkaloids present in endophyte-infected tall fescue (TFS). Additionally, capsaicin-induced vasodilation and heat dissipation may provide a protective effect against hyperthermia, a major issue for cattle consuming TFS [9,11].

We hypothesized that the addition of vasoactive compounds like menthol and capsaicin to molasses-based block supplements could be useful for mitigating vascular responses to ergot alkaloids, such as vasoconstriction, thereby enhancing peripheral blood flow, alleviating heat stress, reducing feed intake depression, and minimizing body weight losses. Menthol acts as an agonist for transient receptor potential cation channel subfamily Melastatin-related member 8 (TRPM8) and capsaicin activates transient receptor potential vanilloid 1 (TRPV1) receptors, which stimulate endothelial nitric oxide synthase (eNOS), leading to nitric oxide (NO) production and vasodilation [10,11]. Overall, the objective of this novel study is to address the knowledge gap in the use of MBS supplementation with or without vasoactive compounds (menthol or capsaicin) on mitigating ergot alkaloid intake’s negative effects on blood perfusion and growth performance.

## 2. Materials and Methods

### 2.1. Experimental Design

The study trial was conducted from 28 July 2018 to 19 October 2018 and was designed as a randomized block design with five treatments using 100 crossbred yearling steers (287 ± 10.35 kg) housed in individual indoor feeding pens. The steers were selected from a larger population of 121 animals, excluding animals with known health issues, poor temperament, or excessively heavy or light body weight. The selected candidates were then stratified based on their initial body weight and assigned randomly, within strata (i.e., blocks), to one of five treatments. Cattle in all treatments had *ad libitum* access to salt blocks and ground prairie hay, referred to as the common diet (Table 1).

The treatments consisted of (1) a negative control (NC), in which animals received only the common diet; (2) a positive control (PC), in which the steers were fed the common diet and a daily aliquot of endophyte-infected tall fescue seed (TFS; Table 1 and Table 2); (3) a control block (CB) treatment consisting of the common diet and *ad libitum* access to a molasses-based block supplement containing approximately 35% crude protein; (4) a similarly formulated block that also contained 0.3% crystalline menthol (MB; manufactured by New Generation Supplements, Belle Fourche, SD, USA); and (5) a similarly formulated block that contained a proprietary blend of mannan oligosaccharide and capsaicin (AB; manufactured by New Generation Supplements, Belle Fourche, SD, USA). Cattle in the block supplement treatments were all given *ad libitum* access to prairie hay, salt blocks, and the same daily aliquot of TFS. Tall fescue seed was presented to the cattle as a mixture with molasses (90% tall fescue seed, 10% molasses), and the amount fed was gradually increased to avoid acute ergot toxicosis, starting with 45 g/animal daily on day 1 of this study and increasing progressively at 3-day intervals to achieve an intake of 520 g/animal daily (Ergosine, 8.75 mg; Ergotamine, 4.94 mg; Ergocornine, 2.13 mg; Ergocryptine, 3.56 mg; Ergocristine, 2.71 mg; Ergovaline, 0.94 mg; the total ergot alkaloid intake was 23.03 mg) by day 42.

The animals were housed in five barns, each containing 20 individual feeding pens, where each barn had 4 individuals from each treatment. The pens were 1.5 m wide by 7.0 m long, constructed of pipe fencing, and open on the south fence. Automatic waterers were located approximately 3.5 m from the feed bunks and were shared by animals in adjacent pens. The feed bunks were situated at the front of each pen.

### 2.2. Cattle Processing and Data Collection

One hundred and twenty-one crossbred steers were purchased in Joplin, Missouri, and transported to the Kansas State University Beef Cattle Research Center. The animals were placed in holding pens and provided *ad libitum* access to ground alfalfa and water on arrival. Due to extremely hot weather, initial processing was delayed by 96 h to avoid excessive handling stress. For initial processing, the steers were individually weighed, identified with numbered ear tags in the right ear, vaccinated against clostridial (Ultrabac7 Somnubac, Zoetis Animal Health) and viral (Bovishield Gold 5, Zoetis Animal Health) pathogens, treated for internal (Safeguard, Merck Animal Health, De Soto, KS, USA) and external (Standguard, Elanco Animal Health, Greenfield, IN, USA) parasites, and implanted with Component TE-200 with Tylan (Elanco Animal Health, Greenfield, IN, USA). The cattle were then placed into their designated treatment pens. 

The steers were presumed to be naive to MBS and thus were given *ad libitum* access as a group to hay and the control molasses-based block for one week prior to initiating the study to acclimate animals to use of the block supplements as part of the adaptation period.

Individual body weights were collected at 21 d intervals starting at day 0 (where the animals were assigned to their treatment groups, thus excluding the 7-day adaptation period from the analyses) and at the end of the 84 d feeding period using an individual animal scale validated immediately before use. Average daily gain was determined by subtracting the previous 21 d interval BW (body weight) from the current BW and dividing the value by days on feed (DOF). The efficiency feed conversion (gain/feed) was determined by taking ADG and dividing it by dry matter intake (DMI). Dry matter intake was determined by adding the intake of the common diet, TFS, and MBS. Infrared thermal imaging data were collected by a trained technician using an FLIR E40 high-performance camera. The temperature range of the camera was −20 °C to 650 °C with a thermal sensitivity of ±0.07 °C and an accuracy of ± 1% of the reading in this restricted range. The wide-angle lens was 25° × 19° (f = 0.4 m). Infrared resolution was 160 × 120 pixels. Emissivity was set to 0.98, with a distance from object of 1 m. All images were collected at 7-day intervals by the same technician. 

Images were collected for each animal from the ocular conjunctiva and the distal ear. If more than one image was taken, the image with the best view of the area of interest was used. These images were then analyzed in the FLIR Tools version 5.13 (Wilsonville, OR, USA) software. The temperature of the ocular conjunctiva (OC) was used as a proxy for core body temperature. The temperature of the distal ear (DE) was determined as the mean of curvilinear points along the periphery of the ear by using the selection tool from the software of FLIR Tools (Wilsonville, OR, USA) thermal imaging analysis, selecting the periphery of the distal third of one ear. The temperature differential (ΔT = OC − DE) was used as an indicator of peripheral blood flow and temperature homeostasis capacity.

Color flow doppler ultrasound imaging was used to identify and measure coccygeal artery diameter. The usage of function color on the ultrasound machines facilitates the identification of blood flow, and it has been widely applied in the livestock reproduction field [12,13,14]. The coccygeal artery was assessed at the 4th coccygeal vertebrae with a high-frequency array transducer (10 MHz, depth of the field was 2.5 cm), a technique reported previously by Klotz et al. (2016) [2]. The first measurement was taken 21 days after the amount of TFS mixture delivered reached its peak. The second measurement was performed 21 days following the first measurement or 42 days after achieving peak consumption of TFS.

Blood samples were harvested on days 63 and 84 from ninety-five animals via jugular vein using 18 G × 25.4 mm needles. One sample from each animal was collected in fluoride oxalate anti-coagulant vacuum tubes for plasma and another in serum tubes (BD Vacutainer, Franklin Lakes, NJ, USA). Plasma and serum were harvested by centrifugation in a swinging bucket rotor at 2000× *g* and 4 °C for 15 min and transferred by pipette to storage tubes, which were then capped and frozen at −20 °C until analysis. Plasma was used to analyze the concentrations (mg/dL) of glucose and L-lactate using a biochemistry analyzer (YSI 2700 STAT Plus, Yellow Springs, OH, USA). Blood serum from day 84 was used with enzyme-linked immunosorbent assays (ELISAs) to quantify the concentrations (ng/mL) of bovine somatotropin (bovine GH-1, MyBiosource, Inc., San Diego, CA, USA) and bovine insulin-like growth factor-1 (bovine IGF-1, Sigma-Aldrich, Saint Louis, MO, USA).

On days 63 and 84, coccygeal artery diameter (CAD) was measured between the 4th and 5th coccygeal vertebrae by color doppler ultrasonography using a portable ultrasound system (Sonosite 180 Plus, Fujifilm, Bothell, WA, USA) equipped with a linear array 10–5 MHz transducer. This method allows for non-invasive assessment of vascular responses; however, certain limitations must be considered. The accuracy of CAD measurements may be influenced by operator variability, transducer positioning, and the depth of the artery relative to surrounding tissues. Additionally, motion artifacts from animal movement and variability in arterial tone due to stress or environmental factors could introduce measurement inconsistencies. To mitigate these issues, all measurements were performed by a single trained operator, and the animals were restrained to minimize movement during scanning. 

Weather data were recorded daily at a local weather station (KKSMANHA69) located 1.4 miles from the Kansas State University Beef Cattle Research Center, Manhattan, KS. The temperature data collected were used to calculate the average environmental temperature (AET).

### 2.3. Diet Preparation and Delivery

All treatment groups received a daily common diet consisting of *ad libitum* prairie hay (Table 1) and common white salt blocks. Treatments that received TFS (PC, CB, AB, and MB) were initially fed 45 g daily. The amount gradually increased in 3 d intervals thereafter until achieving a daily intake of 520 g/animal, which was accomplished by day 42. Intake of the TFS was progressively increased to avoid acute ergot toxicosis. Due to the high palatability conferred by the molasses addition, the animals administered TFS promptly consumed the TFS as soon as it was delivered. The TFS was mixed with sugarcane molasses in a ratio of 9 parts TFS to 1 part molasses (Table 1). Tall fescue seeds were analyzed for ergopeptine alkaloids and ergovaline, the concentrations of which are shown in Table 2. By d 42 of this study, the amount of ergopeptine alkaloids and ergovaline ingested daily were 23.03 and 0.94 mg/animal, respectively.

Cooked molasses-based blocks were provided *ad libitum* for three treatment groups—CB, AB, and MB—and included a 34% protein block, a 34% protein block containing a proprietary blend of mannan oligosaccharide and capsaicin, and a 34% protein block containing 0.3% crystalline menthol, respectively (Table 3). 

The common diet was weighed for each pen daily and delivered to the steers at approximately 1300 h. Fescue seed was ground through a 0.8 mm screen and mixed with sugarcane molasses 9:1 in a horizontal segmented ribbon mixer for 30 min. The TFS mixture was weighed and placed into labeled containers for each pen daily for top-dressing. Bunk management was visually determined and adjusted daily to allow for *ad libitum* intake. Orts were collected at 7-day intervals. To determine dry matter (DM) content, a subsample of the unconsumed feed was dried for 48 h in a 55 °C oven. Dry matter intake was determined for each 7-day interval and at the end of the experiment as follows: DMI = [(total feed accessible × %DM) − (unconsumed feed × %DM)]/interval length in days. 

### 2.4. Statistical Analyses

All analyses were performed using SAS version 9.4 (SAS Institute Inc., Cary, NC, USA). The PROC MIXED procedure was used for cattle growth performance, MBS intake, CAD, ΔT, plasma glucose concentrations, and plasma L-lactate concentrations. Data for serum somatotropin (GH) and insulin-like growth factor I (IGF-1) were determined to be non-normally distributed when evaluated using Shapiro–Wilk and Cramer–von Mises tests and, thus, were transformed into natural logarithms and analyzed using PROC UNIVARIATE. 

An individual animal was the experimental and observational unit. Statistical models included the fixed effects of treatment, time, and the interaction of treatment by time, and weight block and barn were included as random effects. Treatment effects were declared significant at a level of *p* < 0.05. Least-squares means were compared between the treatment groups using the PDIFF function of SAS.

## 3. Results and Discussion

### 3.1. Molasses-Based Block Intake

The overall treatment effect of molasses-based block daily intakes differed among MBS treatments (*p* < 0.05), presented in Figure 1. The MBS daily intake also differed by week (*p* < 0.05), when the lowest weekly intake was during week 3, with an average daily intake of 0.68 kg, and the highest daily intake of 1.25 kg was during week 7. No evidence of treatment by day interaction was identified (*p* > 0.05). The intakes were greater than expected and may have been impacted by maintaining the animals in continuous close proximity to the blocks and housed individually.

### 3.2. Cattle Growth Performance

The growth performance is summarized in Table 4. There were overall treatment effects for overall ADG, DMI, and gain/feed (*p* < 0.01). The improvements in performance due to the supplementation of cooked molasses-based block supplements are consistent with the observations reported by Greenwood et al. (2000) and Ciriaco et al. (2016) [15,16]. The supplementation of protein to cattle fed low-quality forages has been observed to improve fiber digestibility, increase volatile fatty acid (VFA) production, and improve DMI and ADG, as reported by Beaty et al. (1994) and Olson et al. (1999) [17,18].

No difference was observed among the MBS groups, suggesting that the proprietary blend of mannan oligosaccharide with capsaicin and crystalline menthol compounds did not have a significant effect on cattle growth. Van Bibber-Krueger et al. (2016) reported no impact related to menthol supplemented at varying concentrations, including for the same concentration as that used in this study [19]. Previous studies have shown that capsaicin can improve DMI in beef cattle (Cardozo et al., 2006; Rodríguez-Prado et al., 2012) and also increase appetite in humans [20,21,22]. Contrary to these reports, we did not observe an effect of AB compared to other MBS groups. Interestingly, no difference was detected between NC and PC (*p* > 0.05) on any of the growth performance variables, suggesting that in this study, the administration of TFS did not significantly affect growth performance. This result may be attributed to one or more factors, including the dose and duration of TFS exposure, compensatory mechanisms in the animals, or environmental conditions that moderated the expected negative effects of ergot alkaloids. It is possible that the ergot alkaloid concentration in the TFS was not high enough to elicit a measurable reduction in growth performance metrics during the study period. Alternatively, the timeline of the study (84 days) may not have been long enough to capture cumulative growth performance effects, particularly if cattle exhibited gradual adaptation or compensatory growth. 

### 3.3. Coccygeal Artery Diameter Imaging

No evidence of the effect of measurement day or the interaction of treatment by day was detected (*p* > 0.05). The overall treatment effect detected greater luminal areas (*p* < 0.05) for the CB, AB, and MB groups compared to NC and PC, presented in Table 5. The results suggested that those treatments positively influenced the coccygeal artery diameter but did not differ from each other (*p* > 0.05). While menthol and capsaicin are reportedly capable of causing vasodilation [11,23], the results did not differ from those of CB. In this study, the intake of TFS appeared not to have caused a detectable effect on CAD; hence, NC and PC were not statistically different (*p* > 0.05). The failure to detect differences among the MBS groups could be due to the dose of the vasoactive compounds in MBS and also the period of time (84 days) that those animals were fed the MBS. Further research is warranted to investigate if alternative doses of menthol and capsaicin on a longer time frame could show a significant distinction on CAD measurements.

### 3.4. Thermography

The statistical analyses detected significant evidence of the interaction of treatment by day (*p* < 0.05). A more consistent detection of significant difference among treatment groups was evident for the second half (day 42 and greater) of this study. The MBS treatments had lower averages for ΔT (*p* < 0.05) compared to the NC and PC groups, suggesting that peripheral blood perfusion was enhanced with MBS supplementation, as shown in Figure 2. 

The figure highlights the results from the measurements taken on days 77 and 84, where PC displayed a higher temperature differential compared to NC and MBS. This result suggests that the effect of TFS administration negatively impacted blood perfusion. Menthol and capsaicin are known for their vasoactive effects, which was previously reported by Mccarty et al. (2015), Melanaphy et al. (2016), and Craighead et al. (2017); hence, the lower ΔT results in this study are consistent with that effect [9,11,23]. Fortney et al. (1984) and Nose et al. (2018) reported that plasma composition is trivial for efficient thermoregulation, and the results suggest that protein supplementation could have an effect on thermoregulation [24,25]. In this study, the intake of TFS appears to have caused a negative detectable effect on thermoregulation on days 77 and 84; hence, PC had the highest differential temperature (*p* > 0.05). 

### 3.5. Plasma Glucose and L-Lactate

Plasma L-lactate concentrations did not differ among the treatments for the effect of day, treatment, and treatment by day (*p* > 0.05). Our results suggested that MBS supplementation or TFS intake did not alter plasma L-lactate concentrations (*p* > 0.05).

Plasma L-lactate is a biomarker that is easily sampled and has been attracting interest in bovine medicine due to its potential to predict mortality outcomes and assess diseases, including bovine respiratory disease, also known as BRD [26]. 

Plasma L-lactate can also be used to measure general metabolic stress, gluconeogenesis, muscle metabolism, and lipid metabolism. Greater L-lactate concentrations can occur with metabolic insults, and this biomarker is also related to gastrointestinal tract inflammation and acidosis [27]. 

Another widely assessed biomarker is plasma glucose concentration, though complex inter-relationships between multiple tissues and organs can yield variable outcomes [28]. In our study, an effect was not detected on glucose concentration by day of collection and treatment by day interaction measurements. An overall effect of treatment was detected (*p* < 0.05), where the MB group had greater plasma glucose concentrations among PC and NC (*p* < 0.05) but did not differ among AB and CB (*p* > 0.05), as shown in the data presented in Table 6. 

Van Bibber-Krueger et al. (2016) reported no effect on glucose with menthol supplementation [19]. Interestingly, menthol is said to have an antidiabetogenic effect by regulating insulin expression and enhancing pancreatic β-cells [29]. Overall, the administration of TFS and MBS supplementation did not alter plasma concentrations of L-lactate. On the other hand, an effect of treatment was detected when measuring the plasma concentrations of glucose, where the MB group showed the highest levels of the compound. Noteworthily, these measurements had limitations.

### 3.6. Somatotropin and Insulin-like Growth Factor I

The somatotropin (GH) intra- and inter-assay coefficients of variation (CV) were, respectively, 5.2% and 5.1%. Insulin-like growth factor 1 (IGF-1) had a 3.7% intra-assay CV and a 3.9% inter-assay CV. 

The goal for measuring those blood metabolites is that they are very closely related to growth performance, being directly proportional, meaning that greater concentrations of those two metabolites could have a positive effect on mass growth and cell proliferation [19,30,31]. 

In this study, contrary to the hypothesis, no differences among treatment groups were observed. Serum somatotropin (*p* > 0.05) is shown in Figure 3. Serum IGF-1 concentrations also did not differ among treatments (*p* > 0.05), as shown in Figure 4. 

Our hypothesis was that the supplementation of MBS would provide to the animal a greater amount of energy, protein, and minerals that would up-regulate the synthesis and secretion of IGF-I, which would ultimately increase the growth hormone levels, which was observed in studies by Elsasser et al. (1989) [32]. Van Bibber-Krueger et al. (2016) reported more specifically, related to supplementing crystalline menthol, that no significant difference was found on IGF-1 with the addition of crystalline menthol in a diet [19]. Studies pertaining to capsaicin’s effect on animal growth performance were not found.

## 4. Conclusions

In summary, feeding cattle low-moisture cooked molasses-based block supplements can improve ADG, DMI, and feed conversion when offered in conjunction with low-quality prairie hay with or without TFS, and no evidence of a significant difference was observed among the MBS blocks (CB, AB, and MB). The differences among treatments from day 42 to day 84 in differential temperature (ΔT) point out the potential for thermoregulatory enhancement by using MBS, in particular for days 77 and 84. Also, MBS supplementation displayed greater coccygeal artery diameter compared to NC and PC, suggesting that the supplementation of MBS enhanced peripheral blood perfusion in cattle compared to animals not supplemented with MBS. Further research with alternative doses of ergot alkaloids for a longer period is warranted to address some of this study’s limitations. In addition, different MBS formulations with other vasoactive compounds could also be tested to investigate if there is a benefit on mitigating ergot alkaloids intake.

## Figures and Tables

**Figure 1 animals-15-00717-f001:**
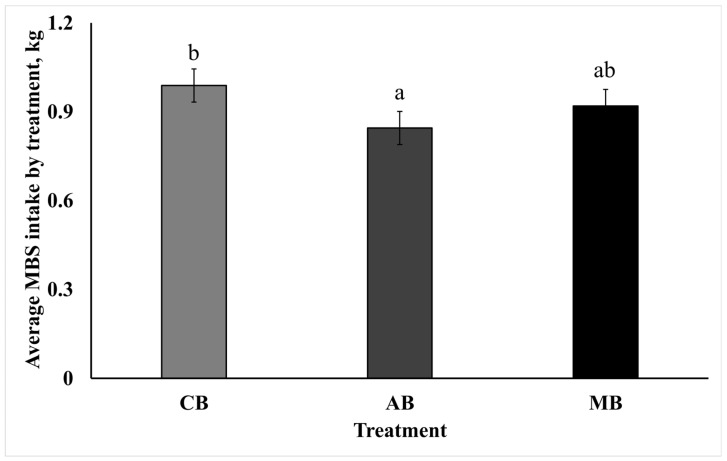
Average daily intake of molasses-based block supplements by treatment. All cattle had *ad libitum* access to ground prairie hay, salt blocks, and their respective block supplements. CB = control block; AB = block containing a proprietary blend of capsaicin and mannan oligosaccharide; MB = block containing 0.3% crystalline menthol. ^a,b^ Means without a common superscript letter display overall treatment effect, *p* < 0.05, SEM = 0.055.

**Figure 2 animals-15-00717-f002:**
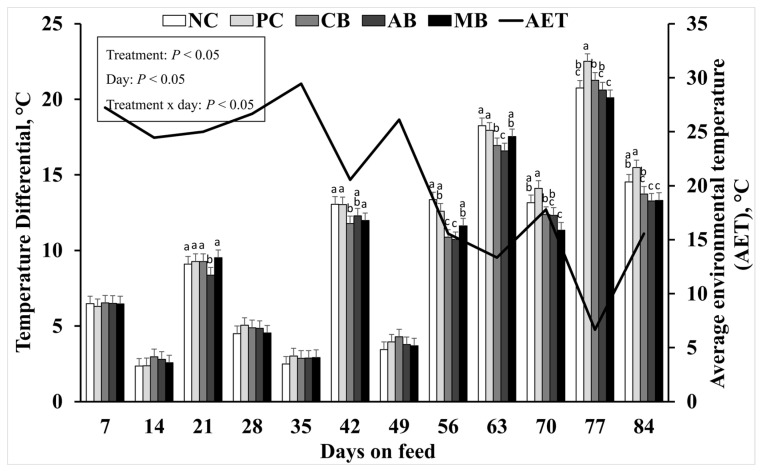
Effect of molasses-based block supplements on temperature differential (ΔT; °C). All cattle had *ad libitum* access to ground prairie hay and salt blocks. NC = negative control treatment with no fescue seed; other treatments were supplemented daily with ergot-infested tall fescue seed (TFS). PC = positive control treatment; CB = control block; AB = block containing a proprietary blend of capsaicin and mannan oligosaccharide; MB = block containing 0.3% crystalline menthol. All block supplements were provided *ad libitum*. ^a,b,c^ Means without a common superscript letter within day are different, treatment × day *p* < 0.05. SEM = 0.5016.

**Figure 3 animals-15-00717-f003:**
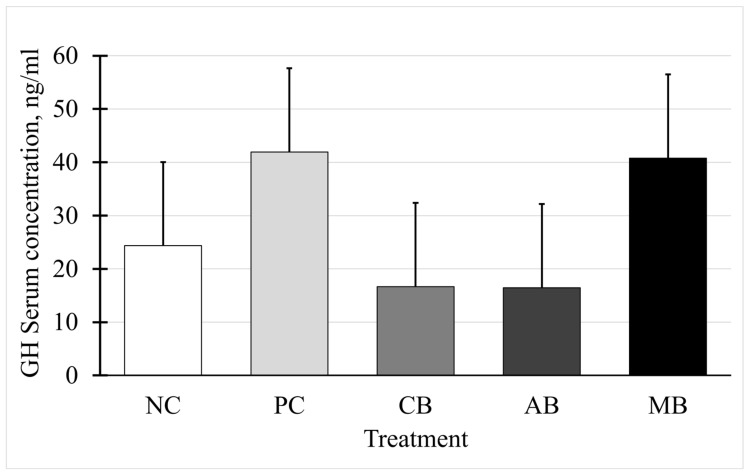
Serum somatotropin (GH) concentrations on day 84. All cattle had *ad libitum* access to ground prairie hay and salt blocks. Treatments are as follows: NC = negative control treatment with no fescue seed; other treatments were supplemented daily with ergot-infested tall fescue seed (TFS). PC = positive control treatment; CB = control block; AB = block containing a proprietary blend of capsaicin and mannan oligosaccharide; MB = block containing 0.3% crystalline menthol. All block supplements were provided *ad libitum*. No treatment effect, *p* = 0.29. SEM = 15.718.

**Figure 4 animals-15-00717-f004:**
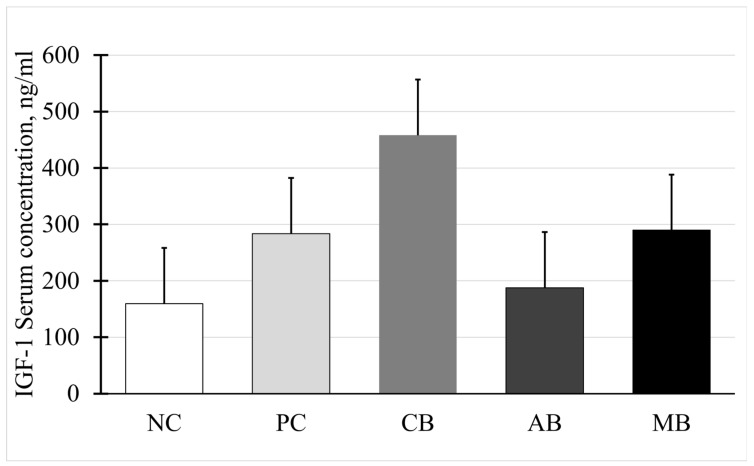
Serum insulin growth factor type 1 (IGF-1) concentrations on day 84. All cattle had *ad libitum* access to ground prairie hay and salt blocks. Treatments are as follows: NC = negative control treatment with no fescue seed; other treatments were supplemented daily with ergot-infested tall fescue seed (TFS). PC = positive control treatment; CB = control block; AB = block containing a proprietary blend of capsaicin and mannan oligosaccharide; MB = block containing 0.3% crystalline menthol. All block supplements were provided *ad libitum*. No effect of treatment, *p* = 0.09. SEM = 98.7.

**Table 1 animals-15-00717-t001:** Composition of basal forage (common diet) and tall fescue seed (TFS) mixture (dry matter basis).

Item, %	Prairie Hay	TFS ^†^
Dry matter	93.09	90.46
Crude protein	5.84	12.60
Acid detergent fiber	47.44	17.02
Neutral detergent fiber	66.52	30.12
Calcium	0.50	0.39
Phosphorus	0.15	0.35
Potassium	1.13	0.18
Magnesium	0.18	-

^†^ A mixture consisting of 90% ergot-infested tall fescue seed and 10% cane molasses.

**Table 2 animals-15-00717-t002:** Ergopeptine and ergovaline contents of tall fescue seed (TFS).

Item	Concentration, Parts per Billion (ppb) ^1^
Ergosine	18,700
Ergotamine	10,560
Ergocornine	4560
Ergocryptine	7600
Ergocristine	5800
Ergovaline	2010
Total	49,230

^1^ The concentration of compounds was measured using high-performance liquid chromatography at a commercial laboratory (Veterinary Diagnostic Laboratory at Missouri State University).

**Table 3 animals-15-00717-t003:** Composition of molasses-based block supplements (dry matter basis).

	Block Supplement ^†^		
Item, %	CB	AB	MB
Dry matter	94.76	94.78	94.53
Crude protein	34.21	34.44	34.35
Crude protein equivalent as non-protein nitrogen	15.22	14.75	15.85
Ash	15.10	15.93	15.21
Calcium	1.57	1.61	1.55
Phosphorus	1.00	0.97	0.99
Potassium	3.19	3.13	3.07

^†^ CB = control block; AB = block containing a proprietary blend of capsaicin and mannan oligosaccharide; MB = block containing 0.3% crystalline menthol.

**Table 4 animals-15-00717-t004:** Effects of ergot-infested tall fescue seed and molasses-based block supplements on cattle growth performance.

	Treatment ^†^					SEM	*p*-Value
Item	NC	PC	CB	AB	MB		
Initial BW, kg	286.3	288.1	288.0	286.2	287.5	6.38	0.40
Final BW, kg	284.9 ^a^	287.4 ^a^	326.6 ^b^	325.9 ^b^	328.4 ^b^	9.01	<0.01
ADG, kg	−0.15 ^a^	−0.03 ^a^	0.46 ^b^	0.43 ^b^	0.60 ^b^	0.117	<0.01
DMI, kg/d	4.98 ^a^	5.24 ^a^	7.19 ^b^	7.05 ^b^	7.29 ^b^	0.247	<0.01
gain/feed	−0.017 ^a^	−0.003 ^a^	0.029 ^b^	0.027 ^b^	0.037 ^b^	0.009	<0.01

^†^ All cattle had *ad libitum* access to ground prairie hay and salt block. NC = negative control treatment with no fescue seed; other treatments were supplemented daily with ergot-infested tall fescue seed (TFS). PC = positive control treatment; CB = control block; AB = block containing a proprietary blend of capsaicin and mannan oligosaccharide; MB = block containing 0.3% crystalline menthol. All block supplements were provided *ad libitum*. ^a,b^ Means without a common superscript letter display overall treatment effect, *p* < 0.05.

**Table 5 animals-15-00717-t005:** Effects of ergot-infested tall fescue seed and molasses-based block supplements on coccygeal artery diameter (CAD; cm) measured between the 4th and 5th coccygeal vertebrae.

	Treatment ^†^						*p*-Value		
	NC	PC	CB	AB	MB	SEM	Trt	Day	Trt × Day
Overall	0.299 ^a^	0.317 ^a^	0.326 ^b^	0.338 ^b^	0.329 ^b^	0.009	0.04	0.10	0.67

^†^ All cattle had *ad libitum* access to ground prairie hay and salt blocks. NC = negative control treatment with no fescue seed; other treatments were supplemented daily with ergot-infested tall fescue seed (TFS). PC = positive control treatment; CB = control block; AB = block containing a proprietary blend of capsaicin and mannan oligosaccharide; MB = block containing 0.3% crystalline menthol. All block supplements were provided *ad libitum*. ^a,b^ Means without a common superscript letter display overall treatment effect, *p* < 0.05.

**Table 6 animals-15-00717-t006:** Effects of ergot-infested tall fescue seed and molasses-based block supplements on plasma concentrations of D-glucose (mg/dL).

	Treatment ^†^						*p*-Value		
Item	NC	PC	CB	AB	MB	SEM	Trt	Day	Trt × Day
D-glucose									
Overall	61.05 ^a^	60.90 ^a^	62.63 ^ab^	64.72 ^ab^	66.04 ^b^	2.437	0.04	0.91	0.27

^†^ All cattle had *ad libitum* access to ground prairie hay and salt blocks. NC = negative control treatment with no fescue seed; other treatments were supplemented daily with ergot-infested tall fescue seed (TFS). PC = positive control treatment; CB = control block; AB = block containing a proprietary blend of capsaicin and mannan oligosaccharide; MB = block containing 0.3% crystalline menthol. All block supplements were provided *ad libitum*. ^a,b^ Means without a common superscript letter display overall treatment effect, *p* < 0.05.

## Data Availability

The original contributions presented in this study are included in this article.

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
