# Peer review of "Molasses-Based Block Supplements for Cattle Fed Endophyte-Infected Tall Fescue (Festuca arundinacea) Seed: Effects on Growth Performance, Circulating Biomarkers, Heat Stress, and Coccygeal Artery Diameter"

_animals, 2025, doi:10.3390/ani15050717_

Round 1
Reviewer 1 Report
Comments and Suggestions for Authors
This manuscript presents a study evaluating the effects of molasses-based block supplements (MBS) on cattle consuming endophyte-infected tall fescue (TFS). The authors hypothesize that MBS containing vasoactive compounds such as menthol and capsaicin may mitigate the adverse effects of ergot alkaloids in TFS, which cause vasoconstriction, heat stress, and reduced growth. The study is well-structured, and the experimental design is appropriate. The results suggest that MBS supplementation improved growth performance and blood circulation, making it a potential strategy for managing cattle on TFS pastures. However, several areas of the manuscript need improvement in terms of clarity, statistical analysis, and depth of discussion.
- The introduction provides a solid background, but the link between the role of molasses-based supplements and their potential to alleviate heat stress and vasoconstriction could be more explicitly tied to the literature. Please consider adding a brief overview of why menthol and capsaicin are chosen and their physiological mechanisms.
- While the hypothesis is clearly stated, the rationale for choosing specific additives (menthol and capsaicin) could be expanded to provide a stronger foundation for their selection based on previous studies.
- The method for measuring coccygeal artery diameter using Doppler ultrasound is well-explained. It may be beneficial to include more information on potential limitations or accuracy issues when measuring this parameter.
- The results in terms of average daily gain (ADG) and dry matter intake (DMI) are presented clearly. However, the manuscript should address why no significant differences were observed between the NC and PC groups in terms of growth performance. Was the dose of TFS insufficient to induce the expected negative effects, or did the growth performance impact occur later than the study period?
- The data on plasma glucose and lactate are relevant but seem inconsistent with the findings in other studies. Further explanation of the lack of significant changes in lactate and glucose concentrations despite improved growth would be valuable.
- The discussion could benefit from more direct comparisons to similar studies. For example, how do the results of this study compare with other research on MBS supplementation or menthol/capsaicin use in livestock?
- There are several non-significant findings (e.g., no difference in CAD between MBS formulations), and it would be helpful to provide more hypotheses for why these results were observed. Is there a possibility that the dose or timing of the intervention was not optimal?
- The authors briefly mention the vasoactive effects of menthol and capsaicin. This section could be expanded to include a more in-depth discussion of their biological mechanisms, such as how these compounds might influence nitric oxide levels, which are known to impact vascular function.
- Conclusion, it could also briefly mention the limitations of the study (e.g., lack of significant difference between MBS formulations) and suggestions for future research, such as exploring long-term effects or optimizing MBS formulations.
Author Response
Thank you for taking the time to peer-review this manuscript. The author's appreciate your willingness to provide insightful comments and suggestions to enhance this document.
The introduction provides a solid background, but the link between the role of molasses-based supplements and their potential to alleviate heat stress and vasoconstriction could be more explicitly tied to the literature. Please consider adding a brief overview of why menthol and capsaicin are chosen and their physiological mechanisms.
Thank you for this suggestion. Introduction expanded to address this suggestion “The occurrence of ergot toxicosis (i.e., fescue foot or gangrenous ergotism) in grazing animals has been reported throughout the world [1]. Ergot alkaloid mycotoxins are secondary metabolites of fungi of the genera Claviceps and Epichloe spp. [1–3]. These fungi can infect grazed forages such as tall fescue (Festuca arundinacea), leading to the clinical condition of fescue toxicosis, characterized by reduced blood flow to peripheral tissues. Altered blood flow is the consequence of ergot alkaloid-induced vasoconstriction by adrenergic and serotonin agonistic effects. Accompanying symptoms of toxicosis include abdominal fat necrosis [4]; hyperthermia; decreases in feed intake, body weight gain, and milk production [5]; lameness, ear tip and tail tip necrosis [6]; thus compromising animal well-being and performance. Moreover, the lack of alternative intervention methods besides removing the animals from the source of ergot-alkaloids urges for alternative methods to mitigate this problem.
Menthol is a multi-purpose compound widely used as a flavoring agent, analgesic, antiseptic, and vasoactive agonist [7]. It acts as an agonist to transient receptor potential cation channel subfamily Melastin-related member 8 (TRPM8), a cold-sensitive ion channel involved in thermoregulation and vasodilation [8]. TRPM8 receptors are expressed in various tissues, including arterial smooth muscle, where activation leads to calcium influx and subsequent vasodilation. This effect is partially mediated by an increase in nitric oxide (NO) bioavailability, which facilitates endothelium-dependent relaxation of blood vessels [11]. Menthol has been reported to induce cutaneous vasodilation, suggesting its potential role in improving blood flow under heat-stressed conditions in livestock [11].
Capsaicin, the active compound in chili peppers, binds to transient receptor potential vanilloid 1 (TRPV1) receptors, which play a crucial role in thermoregulation and vascular function [9]. TRPV1 activation by capsaicin has been linked to enhanced expression and activation of endothelial nitric oxide synthase (eNOS), leading to increased nitric oxide production and vasodilation [10]. This mechanism contributes to reduced vascular resistance, which may counteract the vasoconstrictive effects of ergot alkaloids present in endophyte-infected tall fescue (TFS). Additionally, capsaicin-induced vasodilation and heat dissipation may provide a protective effect against hyperthermia, a major issue for cattle consuming TFS [10,11].
We hypothesized that the addition of vasoactive compounds like menthol and capsaicin to molasses-based block supplements could be useful for mitigating vascular responses to ergot alkaloids, such as vasoconstriction, thereby enhancing peripheral blood flow, alleviating heat stress, reducing feed intake depression, and minimizing body weight losses. Menthol acts as an agonist for transient receptor potential cation channel subfamily Melastin-related member 8 (TRPM8) and capsaicin activates transient receptor potential vanilloid 1 (TRPV1) receptor, which stimulate endothelial nitric oxide synthase (eNOS), leading to nitric oxide (NO) production and vasodilation [9,10]. Overall objective of this novel study is to address the knowledge gap around using MBS supplementation with or without vasoactive compounds (menthol or capsaicin) on mitigating ergot alkaloid intake’s negative effects on blood perfusion and growth performance.”
While the hypothesis is clearly stated, the rationale for choosing specific additives (menthol and capsaicin) could be expanded to provide a stronger foundation for their selection based on previous studies.
Thank you for your comments. Text was modified to “We hypothesized that the addition of vasoactive compounds like menthol and capsaicin to molasses-based block supplements could be useful for mitigating vascular responses to ergot alkaloids, such as vasoconstriction, thereby enhancing peripheral blood flow, alleviating heat stress, reducing feed intake depression, and minimizing body weight losses. Menthol acts as an agonist for transient receptor potential cation channel subfamily Melastin-related member 8 (TRPM8) and capsaicin activates transient receptor potential vanilloid 1 (TRPV1) receptor, which stimulate endothelial nitric oxide synthase (eNOS), leading to nitric oxide (NO) production and vasodilation [9,10].”
The method for measuring coccygeal artery diameter using Doppler ultrasound is well-explained. It may be beneficial to include more information on potential limitations or accuracy issues when measuring this parameter.
This is a great suggestion, thanks for your comment. Author’s enhanced the text by adding “ This method allows for non-invasive assessment of vascular responses; however, certain limitations must be considered. The accuracy of CAD measurements may be influenced by operator variability, transducer positioning, and the depth of the artery relative to surrounding tissues. Additionally, motion artifacts from animal movement and variability in arterial tone due to stress or environmental factors could introduce measurement inconsistencies. To mitigate these issues, all measurements were performed by a single trained operator, and animals were restrained to minimize movement during scanning.”
The results in terms of average daily gain (ADG) and dry matter intake (DMI) are presented clearly. However, the manuscript should address why no significant differences were observed between the NC and PC groups in terms of growth performance. Was the dose of TFS insufficient to induce the expected negative effects, or did the growth performance impact occur later than the study period?
Thank you for this valuable observation, author’s agree the need to expand this discussion. Expanded version reads “Interestingly, no difference was detected between NC and PC (P > 0.05) on any of the growth performance variables, suggesting that in this study, the administration of TFS did not significantly affect growth performance. This result may be attributed to one or more factors, including the dose and duration of TFS exposure, compensatory mechanisms in the animals, or environmental conditions that moderated the expected negative effects of ergot alkaloids. It is possible that the ergot alkaloid concentration in the TFS was not high enough to elicit a measurable reduction in growth performance metrics during the study period. Alternatively, the timeline of the study (84 days) may not have been long enough to capture cumulative growth performance effects, particularly if cattle exhibited gradual adaptation or compensatory growth.”
The data on plasma glucose and lactate are relevant but seem inconsistent with the findings in other studies. Further explanation of the lack of significant changes in lactate and glucose concentrations despite improved growth would be valuable.
Thank you for this observation. We agree to expand this discussion to address this concern.
The discussion could benefit from more direct comparisons to similar studies. For example, how do the results of this study compare with other research on MBS supplementation or menthol/capsaicin use in livestock?
Thank you for your comment. The author’s believe that the relevant studies pertaining the use of menthol and capsaicin for the studied population are referenced. The addition of these compounds to MBS is novel, and are not reported before.
- There are several non-significant findings (e.g., no difference in CAD between MBS formulations), and it would be helpful to provide more hypotheses for why these results were observed. Is there a possibility that the dose or timing of the intervention was not optimal?
Thank you for your suggestion. Text expanded to “The failure to detect differences among MBS groups could be due to dose of the vasoactive compounds in the MBS, and also the period of time (84 days) that those animals fed on the MBS. Further research is warranted to investigate if alternative doses of menthol and capsaicin on a longer time frame could show a significant distinction on CAD measurements.”
The authors briefly mention the vasoactive effects of menthol and capsaicin. This section could be expanded to include a more in-depth discussion of their biological mechanisms, such as how these compounds might influence nitric oxide levels, which are known to impact vascular function.
Thank you for this suggestion. Expanded version “Menthol is a multi-purpose compound widely used as a flavoring agent, analgesic, antiseptic, and vasoactive agonist [7]. It acts as an agonist to transient receptor potential cation channel subfamily Melastin-related member 8 (TRPM8), a cold-sensitive ion channel involved in thermoregulation and vasodilation [8]. TRPM8 receptors are expressed in various tissues, including arterial smooth muscle, where activation leads to calcium influx and subsequent vasodilation. This effect is partially mediated by an increase in nitric oxide (NO) bioavailability, which facilitates endothelium-dependent relaxation of blood vessels [11]. Menthol has been reported to induce cutaneous vasodilation, suggesting its potential role in improving blood flow under heat-stressed conditions in livestock [11].
Capsaicin, the active compound in chili peppers, binds to transient receptor potential vanilloid 1 (TRPV1) receptors, which play a crucial role in thermoregulation and vascular function [9]. TRPV1 activation by capsaicin has been linked to enhanced expression and activation of endothelial nitric oxide synthase (eNOS), leading to increased nitric oxide production and vasodilation [10]. This mechanism contributes to reduced vascular resistance, which may counteract the vasoconstrictive effects of ergot alkaloids present in endophyte-infected tall fescue (TFS). Additionally, capsaicin-induced vasodilation and heat dissipation may provide a protective effect against hyperthermia, a major issue for cattle consuming TFS [10,11].”
Conclusion, it could also briefly mention the limitations of the study (e.g., lack of significant difference between MBS formulations) and suggestions for future research, such as exploring long-term effects or optimizing MBS formulations.
We appreciate this insightful comment. We modified the text to address this concern, where it reads “In summary, supplementing cattle with low-moisture cooked molasses-based block supplements can improve ADG, DMI, and feed conversion when offered to cattle low quality praire hay with or without TFS, no evidence of significant difference was observed among the MBS blocks (CB, AB, and MB). Differences among treatments from day 42 to day 84 in differential temperature (ΔT) points out the potential for thermoregulatory enhancement by using the MBS, in particular for days 77 and 84. Also, MBS supplementation displayed greater coccygeal artery diameter compared to NC and PC, suggesting that the supplementation of MBS enhanced peripheral blood perfusion in cattle compared to animals not supplemented with MBS. Further research is warranted with higher doses of ergot alkaloids for a longer period is warranted to address some of this study’s limitations. In addition, different MBS formulations with other vasoactive compounds could also be tested to investigate if there is a benefit on mitigating ergot alkaloids intake.”
Reviewer 2 Report
Comments and Suggestions for Authors
The manuscript needs revision. Please refer to comments given in the text of reviewed attached file of the manuscript.

Author Response
Line 26: What is the basic problem that your research focuses on and is done to solve?
Thank you for this comment. Modified the text to address this concern. “Ergot alkaloids present in endophyte-infected tall fescue can cause a series of negative effects in exposed cattle. This study evaluated the effectiveness of molasses-based block supplements (MBS) in alleviating vasoconstriction, that leads to reduced peripheral blood flow, heat stress, and impaired growth performance in cattle.”
Line 36-39: This conclusion is wide and general, please add your specific conclusion from your specific results at the end of abstract.
Thank you for this comment. The author’s agree to enhance the clarity in-text in regards to this concern. “These results suggest that molasses-based block supplementation can help mitigate heat stress, poor growth performance associated with ergot alkaloid consumption, potentially providing a practical nutritional strategy for cattle producers managing cattle exposed to ergot alkaloids.”
Introduction section: What is the basic problem that your research focuses on and is done to solve?
Line 45: It is better to explain about importance and application of animal breeding, especially cattle. For this you can use added sentences and references:
Animal breeding in the economy of a country is considered one of the most important economic branches and is of special importance. Animal breeding is a very profitable job and it is considered as a means of raising the economy of countries (Norouzy et al., 2005). Most of the people of the world are engaged in cattle breeding and use its products. In addition, cattle breeding has an important role. Since the creation of mankind, food and nutrition have been one of the major issues for humans, whether when mankind was living wildly in deserts or now, with the help of technology, it has conquered the infinite space. The issue of food and nutrition has been one of the main issues that have occupied human thought, and even though in the new era, mankind has been able to make significant progress in different stages of his life, still the issue of food and nutrition in human societies is a special priority from an economic and social point of view. Today, the importance of nutrition is such that it is considered one of the important criteria of the level of civilization and progress of any society (Mohammadabadi et al., 2010). Because in the all-round development of a society, the level of mental and physical health of the people of that society is the determining factor of animal breeding.
Mohammadabadi MR, Torabi A, Tahmourespoor M, Baghizadeh A, 2010. Analysis of bovine growth hormone gene polymorphism of local and Holstein cattle breeds in Kerman province of Iran using polymerase chain reaction restriction fragment length. African Journal of Biotechnology 9 (41), 6848-6852.
Norouzy A, Nassiry MR, Shahrody FE, Javadmanesh A, Abadi MRM, Sulimova GE 2005 Identification of bovine leucocyte adhesion deficiency (BLAD) carriers in Holstein and Brown Swiss AI bulls in Iran. Russian Journal of Genetics 41 (12), 1409-1413.
Thank you for your thoughtful comment and for taking the time to carefully assess our manuscript. The authors of this manuscript believe that the focus of this manuscript is ruminant nutrition instead of breeding. Furthermore, the authors do appreciate your suggestion to add breeding bit in the introduction, but we would like to keep the focus on cattle nutrition for this work.
Line 69: Please specify the main knowledge gap that your article has filled in the text.
We appreciate this insightful comment. Text was modified to “Overall objective of this study is to address the knowledge gap around using MBS supplementation with or without vasoactive compounds (menthol or capsaicin) on mitigating ergot alkaloid intake’s negative effects on blood perfusion and growth performance.”
Line 71-73: Please specify in the objective whether your research is being conducted for the first time in the world or is it a continuation of another research?
Thanks for your suggestion, added to the text to reflect the novelty of this study. “Overall objective of this novel study is to address …”
Line 73: What is the superiority of your research compared to other researches?
Thanks for your comment, this research has not being done (Using MBS blocks and vasoactive compounds to mitigate ergot alkaloids effects. Added to the text that this was a novel research.
Line 76-78: It would be better to add the age of used animals
Thanks for your suggestion, added to the text “using 100 crossbred yearling steers (287 ± 10.35 kg)”.
Line 127: did you consider adaptation period? how many days? Please identify in the text of the manuscript.
Thanks for your comment. Text modified to “Steers were presumed to be naive to MBS and thus were given ad libitum access as a group to hay and the control molasses-based block for one week prior to initiating the study to acclimate animals to use of the block supplements as part of the adaptation period.
Individual body weights were collected at 21-d intervals starting at day 0 (where animals were assigned to their treatment groups, thus excluding the 7-day adaptation period from the analyses)”
Line 220: Did you use animal statistical model? It would be better to add animal statistical model and its components in the text of the manuscript.
Thanks for your comment. The statistical model is explained on L213-222. Changes were made to clarify the model. “Individual animal was the experimental and observational unit. Statistical models included the fixed effects of treatment, time, and the interaction of treatment by time, weight block and barn were included as random effects.”
Conclusions section: Most of the conclusion is repeat of results and the last sentence of the conclusion is wide and general, please add your specific conclusion from your specific results
We appreciate this insightful comment. “In summary, supplementing cattle with low-moisture cooked molasses-based block supplements can improve ADG, DMI, and feed conversion when offered to cattle low quality praire hay with or without TFS, no evidence of significant difference was observed among the MBS blocks (CB, AB, and MB). Differences among treatments from day 42 to day 84 in differential temperature (ΔT) points out the potential for thermoregulatory enhancement by using the MBS, in particular for days 77 and 84. Also, MBS supplementation displayed greater coccygeal artery diameter compared to NC and PC, suggesting that the supplementation of MBS enhanced peripheral blood perfusion in cattle compared to animals not supplemented with MBS.”